# Efficient Expression in *Leishmania tarentolae* (LEXSY) of the Receptor-Binding Domain of the SARS-CoV-2 S-Protein and the Acetylcholine-Binding Protein from *Lymnaea stagnalis*

**DOI:** 10.3390/molecules29050943

**Published:** 2024-02-21

**Authors:** Lina Son, Vladimir Kost, Valery Maiorov, Dmitry Sukhov, Polina Arkhangelskaya, Igor Ivanov, Denis Kudryavtsev, Andrei Siniavin, Yuri Utkin, Igor Kasheverov

**Affiliations:** 1Department of Molecular Bases of Neuroimmune Signaling, Shemyakin-Ovchinnikov Institute of Bioorganic Chemistry, Russian Academy of Sciences, 117997 Moscow, Russiachem@arkhangelp.ru (P.A.); chai.mail0@gmail.com (I.I.); kudryavtsev@ibch.ru (D.K.);; 2Ivanovsky Institute of Virology, N.F. Gamaleya National Research Center for Epidemiology and Microbiology, Ministry of Health of the Russian Federation, 123098 Moscow, Russia

**Keywords:** LEXSY, receptor binding domain (RBD), acetylcholine-binding protein (AChBP), recombinant protein, nicotinic acetylcholine receptor (nAChR), glycosylation

## Abstract

*Leishmania tarentolae* (LEXSY) system is an inexpensive and effective expression approach for various research and medical purposes. The stated advantages of this system are the possibility of obtaining the soluble product in the cytoplasm, a high probability of correct protein folding with a full range of post-translational modifications (including uniform glycosylation), and the possibility of expressing multi-subunit proteins. In this paper, a LEXSY expression system has been employed for obtaining the receptor binding domain (RBD) of the spike-protein of the SARS-CoV-2 virus and the homopentameric acetylcholine-binding protein (AChBP) from *Lymnaea stagnalis*. RBD is actively used to obtain antibodies against the virus and in various scientific studies on the molecular mechanisms of the interaction of the virus with host cell targets. AChBP represents an excellent structural model of the ligand-binding extracellular domain of all subtypes of nicotinic acetylcholine receptors (nAChRs). Both products were obtained in a soluble glycosylated form, and their structural and functional characteristics were compared with those previously described.

## 1. Introduction

Two protein products were selected for expression in the protozoan system–receptor binding domain (RBD) of the spike-protein of the SARS-CoV-2 virus and the acetylcholine-binding protein from *Lymnaea stagnalis* (AChBP), which are currently actively used for both scientific and practical purposes. RBD, in light of the recent COVID-19 pandemic, has been actively used as the main immunogen contained in the full-length spike-protein (S-protein) for the development of therapeutic antibodies [1,2], as well as in fundamental structural studies of the virus S-protein interaction mechanism with its main target in host cells–angiotensin converting enzyme 2 (ACE2) (see, for example, [3,4,5]).

For more than 20 years, AChBP served as a main surrogate for the ligand-binding domain of nicotinic acetylcholine receptors (nAChRs) [6] which are among the most studied ion channels involved in neural and neuromuscular transmission, as well as in a variety of cognitive, pain, and inflammatory processes [7,8]. For this reason, various cholinergic ligands, which usually also interact with AChBP, are considered promising therapeutic agents for the treatment of dysfunction in the mentioned processes [9,10,11]. Several AChBPs were discovered to date in various mollusks [12,13,14] and some other invertebrates, e.g., spider *Pardosa pseudoannulata* [15]. Two representatives (from *Lymnaea stagnalis* and *Aplisia californica* mollusks) are most often used in scientific research either in the form of wild-type protein (see, for example, [16,17,18]) or after extensive mutagenesis [19,20]. Spider AChBP is considered to be a close structural mimic of the insect nAChRs, which can be used in crop protection-related research [15] complementing the use of *Lymnaea stagnalis* AChBP Q55R point mutant for the structural studies of insecticides [21]. Application of these two naturally occurring proteins gave numerous crystal structures in complex with various cholinergic ligands, which have made it possible to shed light on the mechanism of nAChR functioning, identifying key amino acid residues that determine the affinity and specificity of different nAChR subtypes, their species selectivity, etc. Despite recent advances in nAChR, including their recent cryo-EM studies [22,23,24,25,26], AChBP remains highly relevant: various AChBP complexes have been deposited in PDB seventeen times since 2020 (and 148 structures overall since the early 2000s). 

It is worth noting that both objects (RBD and *Lymnaea stagnalis* AChBP) have complex tertiary structures, contain post-translational modifications (glycosylation), and AChBP is functionally active only in the pentameric form. To date, genetically engineered RBD and AChBPs are usually produced in yeasts [16,27], insects [1,28], mammalian cells [13,29], or bacteria [30,31]. However, all these expression systems either lack glycosylation (bacterial expression), possess non-uniform glycosylation (mammalian cells), or introduce a type of glycosylation that differs significantly from the mammalian type (insect cells, yeast). At the same time, it is known that correct glycosylation can affect the functioning of the protein. For example, it was shown that protein-bound glycosides influence the binding of some toxins to nAChRs and their models [32,33]. It is worth mentioning that in the works listed above, relatively expensive culture media were used for protein production in mammalian cells, requiring a complicated purification system for the target product.

Besides yeasts, insects, and mammalian cells, one more eukaryotic system exists, namely expression in the protozoa *Leishmania tarentolae* [34], which is effective and relatively inexpensive, but not often used. It allows for the generation of proteins with a mammalian type of glycosylation of strictly one type, making the resulting protein uniformly glycosylated [35]. In this work, we report the method for LEXSY-based production of RBD and *Lymnaea stagnalis* AChBP, including the production of appropriate genetic constructs, the expression of target products secreted into the medium, and their purification via two-step chromatography with confirmation of structure and functional activity.

## 2. Results

### 2.1. LEXSY Production of RBD and Lymnaea stagnalis AChBP

The LEXSY P10 host system (Jena Bioscience, Jena, Germany) was used to express both recombinant proteins. This expression strain is derived from *Leishmania tarentolae*, a species of parasitic protozoa isolated from the African gecko *Tarentola mauritanica*, which is not pathogenic to humans. The final genes’ construction for recombinant expression assumed the secretion of the proteins into the extracellular space due to the presence in the manufacturer’s vector of the corresponding secretory phosphatase signal peptide at the N-terminus as well as a C-terminal His_6_-tag for subsequent affinity chromatography. 

To express RBD, a fragment corresponding to the amino acid sequence of this domain (C^336^-P^527^) was amplified (Figure 1) and cloned into a vector obtained at the XbaI-KpnI restriction sites with preservation of the signal peptide sequence for protein secretion into the extracellular space and His_6_-tag for affinity chromatography. 

Similarly, to express *Lymnaea stagnalis* AChBP, the amplified AChBP gene sequence (Leu^1^-Leu^229^) was cloned at XbaI-KpnI restriction sites (Figure 2) with the signal peptide sequence and His_6_-tag. As a result of the production of proteins in the LEXSY system, it was expected that the final recombinant products, in addition to the His_6_-tag, would be flanked by additional amino acid residues from the restriction sites.

### 2.2. Purification of RBD

The first step of the purification procedure of the 72 h cell culture medium with metal chelate affinity chromatography in a stepped imidazole gradient resulted in the separation of the main product (peak 1) eluting with 350 mM imidazole (Figure 3A). The presence of RBD in 350 mM imidazole eluate as well as after dialysis of the collected peak 1 in 20 mM MES was revealed by the results of SDS-PAGE (Figure 3C, tracks 1 and 1a, respectively). The relative molecular mass of RBD was ~25 kDa. 

Subsequent cation exchange chromatography of the dialyzed peak 1 in the linear gradient of NaCl resulted in the elution of a broad peak (Figure 3B). However, the central fraction of this peak, combined into fraction 2 (Figure 3B), showed a high degree of purification of the target product according to SDS-PAGE (Figure 3C, track 2). It was transferred into the suitable buffer, and the fractions of recombinant RBD (Figure 3C, tracks 3 and 4) were used in further studies. The approximate volumes of the resulting fractions and the amount of recombinant RBD in them at the main steps of purification are presented in Table 1. 

### 2.3. LC-MS Study of the Purified RBD

A mass spectrometric analysis (ESI MS) of the protein gave a molecular weight of 23,708.37 Da (Figure 3D), which is higher than the theoretical mass (22,728.39 Da) of the expected recombinant (L^1^-H^202^) RBD (see Figure 1). To confirm the structure of the purified protein, trypsin digestion and subsequent mass-spectrometry analysis have been utilized. Proteomics analysis of the trypsin digest showed that the protein purified from LEXSY culture medium is indeed RBD of the Spike-protein of the SARS-CoV-2 virus because about 86% of the RBD structure was covered in the MS2 fragmentation spectral matches (Figure 3E). 

However, the N-terminal fragment, which contains the only possible glycosylation site in RBD (at N^343^), was not identified by PEAKS (version 10.3) software, which could be explained by the modification of the respective Asn-residue by glycan. Manual analysis of the RBD tryptic digest revealed a fragment with a mass of 2504.05 Da, exactly corresponding to the peptide SLDCPFGEVFNATR with carbamidomethylated Cys residue, containing a glycan with a mass of 892 Da (Figure 3E). This mass corresponds exactly to three hexoses and two N-acetylhexosamines. 

Thus, the expression in the LEXSY system, cultivation, isolation, and purification procedures we developed and performed allowed us to obtain a recombinant glycosylated RBD variant (S^1^-H^203^) of the S-protein of the SARS-CoV-2 virus (Wuhan strain) bearing (Hex)_3_(HexNAc)_2_ glycan at N^11^ with a yield of around 2 mg/L of culture and a purity of not less than 95%.

### 2.4. Testing the Recombinant RBD for SARS-CoV-2 Antiviral Effects

We examined the antiviral activity of the obtained RBD against the SARS-CoV-2 Wuhan-like variant (B.1.1) by its ability to inhibit the virus-induced cytopathic effect (CPE) on Vero E6 cells. It was found that the protein has a pronounced antiviral effect, dose-dependently inhibiting the CPE of the virus with an IC_50_ value of 4.6 µg/mL (Figure 4).

### 2.5. Purification of AChBP

Cell culture medium after 72 h of cultivation subjected to the first step of the purification procedure with metal-chelate affinity chromatography in a stepped imidazole gradient resulted in the separation of the AChBP eluting with 350 mM imidazole (Figure 5A; peak 1). The presence of target protein in the 350 mM imidazole eluate was revealed by the results of SDS-PAGE (Figure 5C, track 1). 

Subsequent gel filtration of the collected peak 1 using the Toyopearl HW-55F resint (Tosoh Corporation, Tokyo, Japan) resulted in effective separation of the recombinant AChBP from the most impurities (Figure 5B). The target protein was eluted as a major symmetric peak around 65–75 min after injection (Figure 5B, fractions 2 and 3), which approximately corresponded to the molecular weight of the pentameric form of the AChBP (~120–130 kDa) according to the elution times of BSA (132 kDa dimer- and 66 kDa monomer-fractions) under the same chromatography conditions (Figure 5B).

SDS-PAGE confirmed the presence of the highly purified AChBP monomer in the collected and concentrated Centricon plus-20 (Millipore, Bedford, MA, USA) fractions 2 and 3 (Figure 5C, tracks 2 and 3, respectively). These SDS-PAGE data were analyzed densitometrically by ImageJ (version 1.54f, NIH, Bethesda, MD, USA) (Appendix A) to calculate the approximate molecular mass of purified AChBP relative to the masses of marker proteins (Appendix A). The calculated apparent molecular mass of the protein was 29.7 kDa, which is higher than the theoretical mass (24.8 kDa). The approximate volumes of the resulting fractions and the amount of recombinant AChBP in them at the main steps of purification are presented in Table 2.

Thus, the cultivation, isolation, and purification procedures we developed and carried out allowed us to obtain recombinant glycosylated *Lymnaea stagnalis* AChBP with a yield of up to 1.5 mg/L of culture and a purity of around 95%. To study possible glycosylation of the purified protein, liquid chromatography and mass-spectrometry (LC-MS) have been performed.

### 2.6. LC-MS Study of the Purified Lymnaea stagnalis AChBP

To confirm the structure of the purified protein, thermolysin digestion and subsequent mass-spectrometry analysis have been utilized. Thermolysin has been used as an enzyme of choice because it produces shorter peptides. Longer tryptic peptides with glycans would have molecular masses out of the dynamic range of the Orbitrap Velos mass-spectrometer, whereas thermolysin-generated glycopeptides should have the optimal molecular masses. To study if the purified *Lymnaea stagnalis* AChBP has glycosylation, the proteolytic peptides were treated with PNGase, which cleaves sugars from the glycopeptide substrates. A comparison of the HPLC profiles of thermolysin-digested AChBP before and after PNGase treatment is presented in Figure 6A and Figure 6B, respectively.

Proteomics analysis of the thermolysin digest showed that the protein purified from LEXSY culture medium is indeed *Lymnaea stagnalis* AChBP (Figure 7A). However, automatic assignment of post-translational modifications has not shown the fragments containing the putative glycosylation site (Figure 7A, arrow) that could be explained by modification of the N-glycosylation site by glycan. A peptide bearing the putative glycosylation site was identified by PEAKS (version 10.3) software in a PNGase-treated thermolysin digest. The respective peptide sequence was assigned to the MS2 fragmentation spectrum (Figure 7B).

To explore the glycosylation of the site, Fragpipe software (version 0.0.0.1, https://github.com/Nesvilab/FragPipe, accessed on 14 February 2024) was used for analysis of the thermolysin digest of AChBP. It was found that the spectra of digests non-treated with PNGase contained *m*/*z* signals in good agreement with peptide ^63^LAWNSSHSPDQVSVP^77^ bearing the additional moiety of 730.3 Da that corresponds to the molecular weight of glycan consisting of two hexose and two N-acetylhexosamines (Figure 8).

As a result, it can be stated that LEXSY-produced *Lymnaea stagnalis* AChBP is N-glycosylated at the N^66^ residue. Thus, at least one post-translational modification consistent with core glycan can be detected in good agreement with previously published data on the LEXSY expression system [35].

### 2.7. Radioligand Binding Assay

To compare the biological activity of the LEXSY-produced protein with the known recombinant *L. stagnalis* AChBP expressed in the insect Sf9 cells, a radioligand binding assay was performed. Radioiodinated α-bungarotoxin ([^125^I]-αBgt) is specifically bound with high affinity (K_d_ 50 ± 14 pM, see Figure 9A) to the *L. stagnalis* AChBP produced in LEXSY. At the same time, a standard specimen of insect cell-produced AChBP has demonstrated nearly identical affinity to [^125^I]-αBgt (K_d_ 62 ± 13 pM, see Figure 9B). These results confirm that the protein contains active binding sites capable of binding snake venom long-type α-neurotoxins. Since such high affinity binding sites are formed at the interface between the AChBP protomers, it can be concluded that a significant part of the expressed protein forms functionally active pentamers.

## 3. Discussion

As already mentioned in the Introduction, both RBD and AChBP are actively used proteins for both scientific and practical purposes, so there is a large amount of data on their expression in various systems. Previously, RBD and AChBPs were expressed in mammalian cells, bacteria, insect cells, and yeast [1,13,16,27,29,30,31,36,37,38].

Each of these expression systems has its drawbacks, while LEXSY combines the qualities of the eukaryotic and bacterial systems with native folding of the expressed protein, a post-translational modification profile, and ease of cultivation. The cultivation of *Leishmania tarentolae* is much cheaper and faster than the cultivation of mammalian cells. In addition, transgenic strains of LEXSY are more stable. One of the important advantages of this system is the possibility of obtaining recombinant products with glycan uniformity. An additional advantageous application of the LEXSY expression system is the possibility to enrich proteins with stable isotopes, enabling structural studies. It can be achieved by growing cells on either complex medium or defined medium supplemented with 15N labeled amino acids essential for the continuous growth of *Leishmania tarentolae* such as Arg, His, Trp, Phe, Ser, Tyr, Thr, Val, Leu, and Lys [39,40]. In fact, the only limitation of the LEXSY expression system is its relatively rare use, resulting in a significantly smaller range of different media and specialized reagents for this system on the market, facilitating cultures of high density and the purification of recombinant proteins free from animal-derived components [41]. The version of the LEXSY expression system used in this work made it possible in both cases to obtain the target recombinant products in soluble form in the extracellular medium with minimal possible differences in the amino acid sequences from the natural ones (Figure 1 and Figure 2).

The protocol developed in this work for the isolation of target products from the extracellular medium was reduced to two-stage chromatography, the first of which was metal-chelate affinity chromatography. It alone made it possible to isolate sufficiently pure target products (see Figure 3A,C (track 1) and Figure 5A,C (track 1)) from a multicomponent extracellular medium. Final purification of the target proteins was carried out using cation exchange chromatography in the case of RBD (Figure 3B) and size exclusion chromatography in the case of AChBP (Figure 5B). Despite a wide eluting peak of RBD during ion-exchange chromatography, its central fractions (after desalination via dialysis) revealed a protein with a relative molecular mass of ~25 kDa and a sufficiently high degree of purification, which was visually assessed by electrophoresis as at least 95% (Figure 3C, tracks 3–4). The measured molecular mass (by ESI mass spectrometry; ESI MS) of the final protein was ~23.7 kDa (Figure 3D), which differs from the calculated mass of the expectable amino acid sequence by ~978 Da. It is known that this fragment of RBD (C^336^-P^527^) contains one glycosylation site at N^343^ in the native SARS-CoV-2 S-protein [42]. Thus, the recombinant RBD obtained by us in the LEXSY expression system probably carries a glycan at this site.

Trypsinolysis accompanied by ESI MS fragments’ analysis showed almost complete coverage of the RBD sequence of the SARS-CoV-2 virus S-protein (more than 86%) (Figure 3E, blue bars), however, PEAKS software could not identify the N-terminal fragment, probably due to the presence of a glycan. Manual analysis of the molecular masses of tryptic peptides revealed a fragment with a mass of 2504.05 Da, exactly corresponding to the mass of the expected N-terminal fragment, which is prolonged with the last residue of the signal peptide (Ser residue), has modified cysteine residue, as well as bears a core glycan of three hexoses and two N-acetylhexosamines (Figure 3E, green bar). It is known, that when in the LEXSY system the signal peptide is removed during the processing of expressed protein, an extended or shortened N-terminal version of the target product may be formed. The SignalIP 5.0 DTU Health Tech online server predicted around 0.7 (signal peptidase cleavage site at VDA-GASLD) and 0.3 (signal peptidase cleavage site at VDAGA-SLD) probabilities of having an additional GAS-triplet or single S-residue, respectively, at the N terminus of the expected recombinant RBD after the signal peptide was removed.

The naturally-occurring S-protein of the SARS-CoV-2 virus is a heavily glycosylated molecule, for which 22 glycosylation sites have been established [42]. However, the RBD domain (the variant that was expressed in this study–C^336^-P^527^) contains the only glycosylation site at N^343^. A glycan with a complex-type structure was revealed in a natural protein that is produced in host cells [42]. A glycan with a very similar composition is also produced by the expression of the S-protein (or RBD alone) in commonly used mammalian cell lines (HEK and others) [43]. Expression in yeast or insect cells yields a simpler (but also not uniformed) composition of glycans, mainly of the mannose-type [1,27,44]. As established in this work, in the recombinant product we obtained in the LEXSY system, the structure of the oligosaccharide bound to the same asparagine residue is even simpler and represents a homogeneous core glycan–(Hex)_3_(HexNAc)_2_.

Size exclusion chromatography of purified *Lymnaea stagnalis* AChBP on a metal-chelated column showed that the main peak, according to the calibration of standard proteins, elutes in the region of ~120–130 kDa, which corresponds to the pentameric form of this protein, which is a naturally occurring functionally active form (Figure 5B). At the same time, the observed relative molecular mass of the AChBP monomer according to SDS-PAGE was 29.7 kDa (Figure 5C, tracks 2–3), which is higher than the theoretical mass of 24.8 kDa of the expectable polypeptide chain without a signal peptide. *Lymnaea stagnalis* AChBP expressed in *E. coli* has been reported to migrate on SDS-PAGE gel with an apparent mass of 19.2 kDa [30]. Since *E. coli* lacks the glycosylation apparatus in contrast to LEXSY, the discrepancy between the predicted and apparent molecular masses might be attributed to the glycosylation that is known to alter the electrophoretic mobility of the protein [45].

To confirm the structure of the recombinant product and the presence of a glycosylation site, as in the case of RBD, ESI mass spectrometry analysis was applied for its proteolytic fragments, but this time thermolysin was used to obtain shorter peptides. Similarly, almost complete coverage of the sequence of *Lymnaea stagnalis* AChBP (86.2%) was also obtained (Figure 7A, blue bars). However, as in the case of RBD, PEAKS software was unable to identify a fragment (^63^LAWN^66^) potentially bearing a glycosylation site (Figure 7A, arrow), probably due to the presence of glycan. It is known that the only possible N-linked glycosylation site of the naturally occurring *Lymnaea stagnalis* AChBP is located just at the N^66^ residue [12]. At the same time, the presence of a glycosylated fragment was confirmed by comparing the HPLC elution profiles of fragments of thermolysin-digested AChBP before and after treatment with PNGase, as shown by the disappearance of one and the appearance of two new peaks in the corresponding profiles (Figure 6A,B). Deglycosylated fragments containing a glycosylation site were detected by PEAKS software, and MS2 fragmentation confirmed the structure of one of them (Figure 7B).

On the other hand, the application of Fragpipe software to an untreated PNGase fragment, presumably bearing a glycan (Figure 6A, red arrow), showed a molecular mass of ~2354 Da (Figure 8A,B), and its MS2 fragmentation revealed the structure ^63^LAWNSSHSPDQVSVP^77^ (Figure 8C). The difference in molecular mass was 730.3 Da, which corresponds to core glycan consisting of two hexose, and two N-acetylhexosamines (Figure 8C).

As a result, the LEXSY expression system made it possible to quickly and relatively inexpensively obtain glycosylated recombinant proteins–RBD of the SARS-CoV-2 virus S-protein and *Lymnaea stagnalis* AChBP in a soluble form in extracellular medium. Yields ranged from about 1.5 to 2.0 mg/L of culture for highly purified target products (with a purity of at least 95%) (Table 1 and Table 2). These values roughly correspond to the yields of the same proteins (in the case of calculation for highly purified products) expressed in insect or mammalian cells: 0.5–2.5 mg/L of culture for different AChBPs [13,15,46,47] and 0.82 mg/L of culture for RBD [48], but slightly lower for *E. coli*-recombinants: 1–1.5 mg/L of culture for *A. californica* AChBP and 4–5 mg/L of culture for *L. stagnalis* AChBP [30]. It is worth noting that often published values of RBD yields of tens and even hundreds mg/L of culture (see, for example, [27,29]), obtained in different expression systems, are not estimated based on the final highly purified product.

We determined that the recombinant RBD expressed in the LEXSY system has antiviral activity against the SARS-CoV-2 virus (IC_50_ 4.6 µg/mL), which is close to the activity of the protein expressed in *E. coli* [31]. Previously, it was shown that RBD protein expressed in 293T cells inhibits SARS-CoV-2 pseudovirus entry in a dose-dependent manner with an IC_50_ of 1.35 μg/mL [49]. Lipopeptides targeting the HR1 and HR2 domains of the S2 subunit of the SARS-CoV-2 S protein demonstrated potent inhibitory activity against SARS-CoV-2 pseudoviral infection with nanomolar IC_50_ values [50]. The trimeric RBD version of the SARS-CoV-2 spike protein blocked SARS-CoV-2 binding to ACE2, thereby blocking viral infection (IC_50_ was 1.6 μg/mL) [48]. Moreover, the RBD protein fused to the Fc domain of human IgG demonstrated potent antiviral efficacy and also inhibited the attachment of SARS-COV-2 to mouse lungs [51]. These results indicate that the recombinant RBD proteins (including those expressed in LEXSY) can be used to prevent and treat SARS-CoV-2 infection. The need to obtain a glycosylated form of RBD is currently debatable. On the one hand, a fairly large number of studies have been published on the *E. coli* expressed RBD, which not only made it possible to obtain effective antibodies against the SARS-CoV-2 virus but also demonstrated its direct interaction with ACE2 with an efficiency close to the deglycosylated product (see, for example, [31,52,53]). It has also been shown that de-glycosylation of recombinant RBD samples expressed in mammalian cells does not significantly reduce the effectiveness of their interaction with ACE2 [54].

On the other hand, a number of studies support the important role of glycosylation. It has been shown that RBD obtained in different expression systems has different immunogenicity in animals [55], which may be important in the design of prototype vaccines. Thus, studies of a prototype RBD-based vaccine have shown that immunization with glycosylated protein induces enhanced neutralizing antibodies against SARS-CoV-2 [56]. A number of papers have also been published on the importance of glycosylation of RBD in the study of immunogenicity, affinity for ACE2, and infectivity of the virus [44,57,58].

In addition, a large number of studies have recently appeared indicating the presence of other additional biological targets for recognition by the SARS-CoV-2 S-protein (and, in particular, its fragment–RBD), with the different nAChR subtypes being increasingly mentioned among them (see, for example, [59]). At the moment, the role of S-protein glycosylation in this interaction has not been studied at all. These considerations argue in favor of obtaining glycosylated RBD samples for research and thus are in favor of using the LEXSY expression system.

As to the AChBPs, the role of their glycosylation and glycans’ composition is practically unresolved at the moment. The oligosaccharide moiety has not been characterized for any of the currently known naturally-occurring AChBPs. Discovered in glial tissue, the first AChBP from *Lymnaea stagnalis* (Ls-AChBP) has shown the ability to effectively interact with acetylcholine and α-bungarotoxin [12], but its structural X-ray studies, as well as detailed ligand-binding properties, have only been studied on recombinant proteins, obviously with non-native located at N^66^ glycan obtained in various expression systems [16,30,60,61]. This is also true for other DNA-derived AChBPs expressed in various systems [13,14,15,17,62]. It is worth noting that in most of the works, recombinant AChBPs were obtained as functional and structural homologues of nAChRs for crystallographic studies of the spatial organization of ligand-binding sites, while studies of their glycan moiety remained on the sidelines.

Meanwhile, glycosylated AChBPs have also been successfully produced in yeast, insect, and mammalian cells. However, the latter often results in highly N-glycosylated and nonhomogeneous proteins, which might prevent successful crystallization [13]. This problem was solved by expressing them in glycosylation-deficient mammalian cell lines to obtain homogeneous AChBPs with shorter glycan chains [47], for which numerous X-ray structures of their complexes with various ligands were obtained.

In some cases, crystallographic studies have been able to resolve the structure of oligosaccharides in AChBPs and their chimeric variants, typically presented as a core glycan ((Hex)_3_(HexNAc)_2_) or parts of it (see, for example, the X-ray structures with PDB ID–2pgz, 2wn9, 5oui, and 5o8t and the corresponding references [63,64,65,66]). At the same time, the uniformity of glycans in protein protomers is often absent. The product obtained in the LEXSY system in this study avoided this problem by giving a similar uniform core glycan (Hex)_2_(HexNAc)_2_ (Figure 8C). We believe that this may simplify the production of crystals of this product with new ligands, including compounds with moderate affinity.

One of the few published works devoted to the study of the AChBP oligosaccharide profile was the work [62] carried out on the protein from the marine annelid *Capitella teleta*. Interestingly, its expression in mammalian cells revealed a wide variety of complex-type glycans located at N^122^ and N^216^, far from the glycosylation sites of *Lymnaea stagnalis* and *Aplysia californica* AChBPs (N^66^ and N^74^, respectively).

The recombinant *Lymnaea stagnalis* AChBP expressed in the LEXSY system obtained in this work showed complete identity because of its high affinity to radiolabeled α-bungarotoxin, as compared with the same protein expressed in Sf9 insect cells (K_d_ 0.050 vs. 0.062 nM, respectively) (Figure 9). The last one (together with *Aplysia californica* AChBP) was obtained in the laboratory of Prof. Sulan Luo [67]; these proteins were earlier successfully used by us in joint studies on the characterization of various nAChR subtypes using conopeptides [68,69]. α-Bungarotoxin from the venom of the multi-banded krait *Bungarus multicinctus* belongs to the three-finger long-type α-neurotoxins, which constitute one of the most prominent groups in the venom of snakes (kraits and cobras), and for almost 60 years α-neurotoxins have been serving as one of the most versatile tools for the study of distinct subtypes of nAChRs (see, for example, reviews [70,71]), and in the last 20 years also of AChBPs [6,12,17].

The obtained similar affinities for the two *Lymnaea stagnalis* AChBPs expressed in insect and leishmania cells (the greatest difference of which probably lies in the carbohydrate moiety) were expected. Despite the lack of data on the activity of naturally occurring AChBP towards cholinergic ligands, certain conclusions can be drawn at least about the role of posttranslational modifications (including the role of glycans) in the binding properties of this protein, since it was possible to develop a technique for its expression in *E. coli* [30]. It was shown that the *E. coli*-expressed *Lymnaea stagnalis* and *Aplysia californica* AChBPs retain similar affinity for a number of tested ligands (epibatidine, nicotine, and α-conotoxin ImI) with glycosylated proteins expressed in insect cells. This suggests that glycosylation has no effect on ligand binding to these AChBPs, which is not surprising since their glycosylation sites (see above) are located far from the binding sites for cholinergic agonists and competitive antagonists.

This fact contrasts quite strongly with the situation with respect to naturally occurring nAChRs, for which much more data has been obtained on the role of glycosylation in receptor activity. In particular, the glycosylation sites of muscle-type nAChR subunits from the fish electrical organ and the complex composition of glycans, for example, at N^141^ of the α-subunit located close to the functionally important C- and Cys-loops, are precisely known [72]. Therefore, it is not surprising that a fairly large amount of data has been published indicating the important role of glycans in folding, assembly, and trafficking of nAChRs, as well as the efficient binding of some ligands (see, for example, the review [73]). In contrast, the functional role of glycans in AChBPs remains to be elucidated.

One of the approaches aimed at obtaining an improved model of the ligand-binding domain of different nAChRs was the creation of recombinant chimeric proteins based on their structures, with elements from AChBP included. Many of these chimeric homologues showed pharmacological profiles close to those of natural nAChRs and were successfully crystallized [19,65,74]. It is in such structures that the presence of glycans characteristic of natural nAChRs can play an important role [75,76], which increases the importance of glycosylated AChBP expression in the LEXSY system.

## 4. Materials and Methods

### 4.1. LEXSY Production of RBD and Lymnaea stagnalis AChBP

To express RBD, a vector pLEXSY-sat2.1-RBD-His_6_ was obtained. From the initial pVAX-1-S-glycoprotein vector (Evrogen, Moscow, Russia) containing the gene sequence of the full-length S-protein of the SARS-Cov-2 virus (Wuhan strain), a fragment corresponding to the amino acid sequence of RBD (C^336^-P^527^) was amplified (Figure 1). It was cloned into the manufacturer’s pLEXSY-hyg 2.1 vector at the XbaI-KpnI restriction sites with preservation of the signal peptide sequence for protein secretion into the extracellular space. The antibiotic Hygromycin B (AppliChem, Darmstadt, Germany) was used to select transformed clones. Cloning was performed using chemically competent cells of the *E. coli* strain XL1-Blue (Evrogen, Moscow, Russia); the correct gene encoding vector clones were chosen after sequencing (Evrogen, Moscow, Russia).

25 ug of pLEXSY-hyg2.1-RBD-His_6_ vector was linearized with SwaI restriction enzyme (Neb, Ipswich, MA, USA) and precipitated from the enzymatic mixture with ethanol. DNA diluted in nuclease-free water (Evrogen, Moscow, Russia) was transformed into the LEXSY culture, which was washed with electroporation medium according to the manufacturer’s high-voltage protocol (BioRad, Hercules, CA, USA). Further selection of polyclonal culture was carried out with Hygromycin B antibiotic (AppliChem, Germany) for 7–10 days. For protein production and its subsequent purification, the transformed LEXSY strain was cultivated at 26 °C in T225 vented cap flasks (Corning Somerville, MA, USA) in BHI medium (BD Biosciences, San Jose, CA, USA) supplemented with Hemin (500× stock solution of 0.25% Hemin in 30% Triethanolamine) and Penicillin/Streptomycin antibiotics (100× stock solution of 10,000 U/mL Penicillin, 10,000 mg/mL Streptomycin). The medium containing the recombinant product was collected by centrifugation, followed by filtration through PVDF membranes (Millipore, Germany).

To express *Lymnaea stagnalis* AChBP (Leu^1^-Leu^229^), the vector pLEXSY-sat2.1-Ls-AChBP-His_6_ was obtained using the vector pLEXSY-sat2.1 provided by the manufacturer. The amplified AChBP gene sequence was cloned at XbaI-KpnI restriction sites (Figure 2) with preservation of the signal peptide sequence for protein secretion into the extracellular space and His_6_-tag for affinity chromatography. Cloning was performed using chemically competent cells of the *E. coli* strain XL1-Blue (Evrogen, Moscow, Russia); the correct gene encoding vector clones were chosen after sequencing (Evrogen, Moscow, Russia).

Linearization of the pLEXSY-sat2.1-Ls-AChBP-His_6_ vector, its transformation into the LEXSY culture, selection of transformants, and their cultivation with the production of recombinant AChBP in the extracellular space were carried out similarly to the RBD.

### 4.2. RBD Purification

The target protein was isolated from the filtered medium after 72 h cultivation. Purification steps included metal chelate affinity chromatography, dialysis, ion exchange chromatography, and desalting. NaCl and a stock solution of 1 M Tris-HCl buffer, pH 8.0, were added to the final concentration of 100 mM in the collected supernatant. Affinity chromatography was carried out on a C 10/10 column (Pharmacia, Stockholm, Sweden) packed with INDIGO Ni-Agarose resin (Cube Biotech, Monheim am Rhein, Germany). The column was equilibrated with buffer A (100 mM NaCl, 20 mM Tris-HCl buffer, pH 8.0). The elution of RBD was performed in 35% buffer B (1 M imidazole, 100 mM NaCl, 20 mM Tris-HCl buffer, pH 8.0). The collected fraction contained protein of interest, which was supported by 12% Tris-Glycine SDS-PAGE. The protein sample was dialyzed against buffer C (20 mM NaCl, 20 mM MES, pH 5.0) within 48 h with confirmation of the desired product band by SDS-PAGE.

The next step of purification was a cation exchange chromatography on SP Sepharose (Amersham Biosciences, Amersham, UK) resin packed in the empty 1 mL FPLC column (Biocomma, Shenzhen, China) and equilibrated with buffer C (20 mM NaCl, 20 mM MES, pH 5.0). The sample was eluted in a NaCl gradient from 0 to 1 M during 20 min at 1 mL/min flow. Buffer in the collected peak fractions was exchanged to 50 mM Tris-HCl buffer, pH 7.6, by means of PD10 desalting columns (Cytiva, Marlborough, MA, USA); the presence of the target product was checked by 12% Tris-Glycine SDS-PAGE. The concentration of the RBD in fractions was determined by the Bradford assay using bovine serum albumin (BSA) as a standard on a plate reader (Hidex, Turku, Finland) in accordance with the manufacturer’s recommendations (Bio-Rad, USA).

### 4.3. Enzymatic Digestion and Mass-Spectrometry of RBD

15 μg of the purified protein was dissolved in 60 µL of denaturation buffer (100 mM Tris-HCl buffer, pH 8.5 (Helicon, Moscow, Russia), 2% sodium deoxycholate (SDC; Sigma Aldrich, St. Louis, MO, USA), 10 mM 2-chloroacetamide (CAA; Vekton, Moscow, Russia), 10 mM tris(2-carboxyethyl)phosphine (TCEP; Sigma Aldrich, USA)). The sample was incubated at 85 °C for 30 min, after which it was diluted with 3D-water^TM^ to the SDC concentration of 1%. Modified recombinant LC-MS-grade trypsin (Molecta, Moscow, Russia) was added in a 1:100 (enzyme/protein) ratio, and the sample was incubated overnight at 37 °C. Trypsin was deactivated by the addition of 20 μL of 10% trifluoroacetic acid (TFA). The sample was cleaned by liquid-liquid extraction (5 iterations) using water-saturated ethyl acetate (Component-Reaktiv, Moscow, Russia) to remove SDC. The extract was then evaporated in a SpeedVac concentrator, and tryptic peptides were desalted by the STAGE Tips MCX-type protocol. The final peptide solution was dried in a SpeedVac concentrator, and after that, the sample was reconstituted in 5% acetonitrile, containing 0.1% TFA, in water to a total peptide concentration of 7.5 µg/μL.

LC-MS/MS was performed using the Dionex Ultimate 3000 NCS-3500RS chromatographic nano-HPLC system (Thermo Scientific, Waltham, MA, USA) coupled with the Orbitrap Elite mass spectrometer (Thermo Scientific, USA). The peptide samples were initially concentrated on a pre-column (100 μm × 50 mm, packed with Inertsil C18 3 μm sorbent; GL Sciences, Tokyo, Japan) at a flow rate of 3 µL/min in 5% acetonitrile, 0.1% formic acid, and 0.02% TFA in water. The samples were then separated on the analytical nano-column (75 μm × 150 mm, packed with Aeris Peptide C18 1.7 μm sorbent; Phenomenex, Torrance, CA, USA) in an 80-min linear gradient (curve coefficient = 5) of 5% to 55% acetonitrile, 0.1% formic acid, and 0.02% TFA in water at a flow rate of 300 nL/min.

Eluting peptides were analyzed on-line in the Thermo Orbitrap Elite mass spectrometer (Thermo Scientific, San Jose, CA, USA) via nano-ESI (3 kV emitter voltage and 260 °C capillary temperature). The analysis was performed in DDA mode. MS1 scanning was carried out in an orbitrap ion trap in a mass range of 200–2000 *m*/*z*, with a maximum IT of 100 ms and a resolution of 60,000. The 15 most abundant precursor ions were chosen for fragmentation in a collision-activated linear ion trap (CID). MS2 scanning was performed in a linear ion trap, maximum IT 25 ms, isolation window 1 *m*/*z*, NCE 35%. Dynamic exclusion was set to 120 s.

### 4.4. AChBP Purification

The AChBP purification protocol consisted of immobilized metal affinity chromatography, as described for RBD, and size exclusion chromatography. After 72 h cultivation, the supernatant was collected by centrifugation and PVDF membrane filtration. NaCl and 1 M Tris-HCl buffer, pH 8.0, were added to the final concentration of 100 mM. Target protein was captured on a C 10/10 column (Pharmacia, Stockholm, Sweden) packed with INDIGO Ni-Agarose resin (Cube Biotech, Germany) equilibrated with buffer A (100 mM NaCl, 20 mM Tris-HCl buffer, pH 8.0). The elution of AChBP was performed in 35% buffer B (1 M imidazole, 100 mM NaCl, 20 mM Tris-HCl buffer, pH 8.0). The collected fraction contained protein of interest, which was supported by 12% Tris-Glycine SDS-PAGE.

The next step was size exclusion chromatography in 50 mM Tris-HCl buffer, pH 7.6, to isolate the fraction of AChBP pentameric form on a C 16/60 column (GE Healthcare, Chicago, IL, USA) with a Toyopearl HW-55F carrier (Tosoh Corporation, Tokyo, Japan) equilibrated by 50 mM Tris-HCl buffer, pH 7.6. The peak with the appropriate mass for the pentameric product was collected and concentrated to ~0.4 mL using Centricon plus-20 with 10 kDa MWCO (Millipore, Bedford, MA, USA), with confirmation of the desired monomeric product band by 12% Tris-Glycine SDS-PAGE. The concentration of the resulting protein solutions was determined in the same way as in the case of RBD by means of a Bradford assay (BioRad, Hercules, CA, USA).

### 4.5. Enzymatic Digestion and Mass-Spectrometry of AChBP

To 100 micrograms of the purified protein dissolved in digestion buffer (50 mM Tris-HCl buffer, pH 7.6, containing 0.5 mM CaCl_2_), 5 micrograms of thermolysine (Promega Corporation, Madison, WI, USA) have been added according to the manufacturer’s recommendations. The reaction sample was incubated at 95 °C for 1 h, then the digestion process was stopped by the addition of formic acid to the final concentration of 10%.

ESI MS spectra were recorded on an LTQ Orbitrap Velos instrument (Thermo Scientific, San Jose, CA, USA) equipped with an Agilent Nanoflow LC system (Agilent Technologies, Santa Clara, CA, USA). Acetonitrile gradients of 5 to 55% during 70 min (Buffer A: 0.1% formic acid and 0.02% TFA in water; Buffer B: 0.1% formic acid and 0.02% TFA in acetonitrile) were used to achieve peptide fragment separation. Spectra were recorded in FTMS with a positive NSI full ms scan regime in the *m*/*z* range of 300.00–2000.00.

### 4.6. SARS-CoV-2 Antiviral Assay

The biological activity of the recombinant RBD was assessed by its effect on SARS-CoV-2 replication (hCoV-19/Russia/Moscow_PMVL-4, GISAID ID: EPI_ISL_470898), as described previously [77]. Briefly, different protein dilutions were added to a monolayer of Vero E6 cells in 96-well plates. Then, the cells were infected with SARS-CoV-2 at 100 TCID_50_. The virus-induced cytopathic effect (CPE) was assessed using the MTT method at 72 h after infection.

### 4.7. Radioligand Binding Assay

The biological activity of the recombinant *Lymnaea stagnalis* AChBP was assessed by its ability for equilibrium saturation binding of the [^125^I]-labeled α-bungarotoxin ([^125^I]-αBgt). As a comparison control, the same protein obtained in the Bac-to-Bac baculovirus expression system [67] was studied in parallel. The binding of 0.026 μg/mL proteins’ solutions with different concentrations of [^125^I]-αBgt (from 0.08 to 1.32 nM) during 35 min was carried out in 50 mL of binding buffer (20 mM Tris-HCl buffer, pH 8.0, containing 1 mg/mL BSA) at room temperature. Nonspecific binding was determined by preliminary 2.5-h incubation of AChBPs with 5.2 µM α-cobratoxin. AChBP solutions were applied to double DEAE-cellulose paper filters (Whatman, Maidstone, UK) presoaked in washing buffer (20 mM Tris-HCl buffer, pH 8.0, containing 0.1 mg/mL BSA), and unbound radioactivity was removed from the filter by washing (3 × 3 mL) with washing buffer. The bound radioactivity on filters was determined using a Wallac Wizard 1470 Automatic Gamma Counter (PerkinElmer Life and Analytical Sciences, Turku, Finland).

The equilibrium binding data were fitted using ORIGIN 7.5 to a one-site model according to the equation: B(x) = B_max_ /(1 + K_d_/x), where B(x) is the radioligand specifically bound at free concentration x (determined by subtracting the amount of bound and adsorbed radioligand from the total amount added to the incubation mixture), B_max_ is the maximal specific bound radioligand, and K_d_ is the dissociation constant.

## 5. Conclusions

In this work, we showed that by applying the *Leishmania tarentolae* derived LEXSY expression approach, it is possible to quickly and inexpensively obtain two currently actively used proteins–the receptor binding domain of the SARS-CoV-2 Spike-protein and the acetylcholine-binding protein from *Lymnaea stagnalis*. Both products are obtained in soluble form in the extracellular space, which allows for a purity of at least 95% by two-stage chromatography and good yields (1.5–2.0 mg/L of culture). The structures of both proteins modified with certain glycans were confirmed by mass spectrometry, and their biological activity in the tests used does not differ from that of recombinant products obtained in other expression systems.

## Figures and Tables

**Figure 1 molecules-29-00943-f001:**
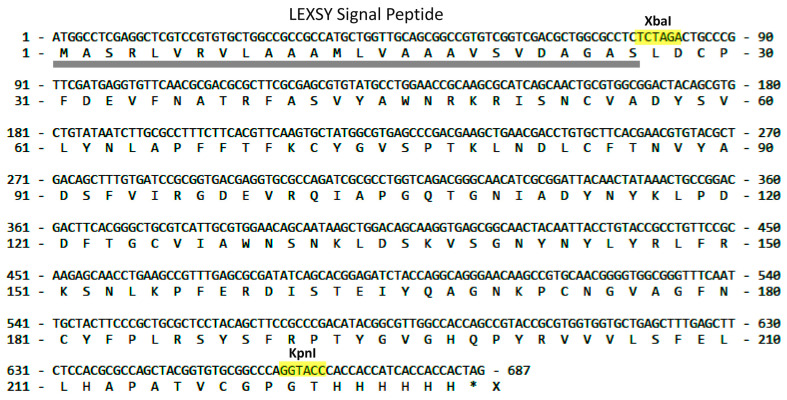
Nucleotide sequence of RBD cloned into the pLEXSY-hyg 2.1 vector. The XbaI and KpnI restriction sites (colored in yellow) belong to the manufacturer’s vector. The secretory expression pathway leads to proteolytic cleavage of the signal peptide (underlined in gray). The asterisk symbol refers to stop codon.

**Figure 2 molecules-29-00943-f002:**
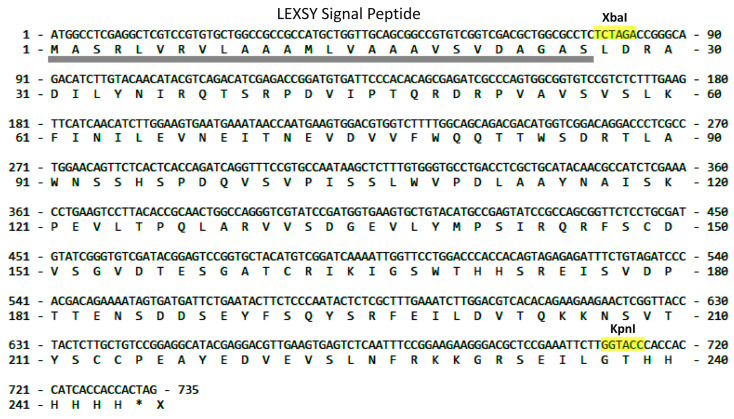
Nucleotide sequence of *Lymnaea stagnalis* AChBP cloned into the pLEXSY-sat2.1 vector. The XbaI and KpnI restriction sites (colored in yellow) belong to the manufacturer’s vector. The secretory expression pathway leads to proteolytic cleavage of the signal peptide (underlined in gray). The asterisk symbol refers to stop codon.

**Figure 3 molecules-29-00943-f003:**
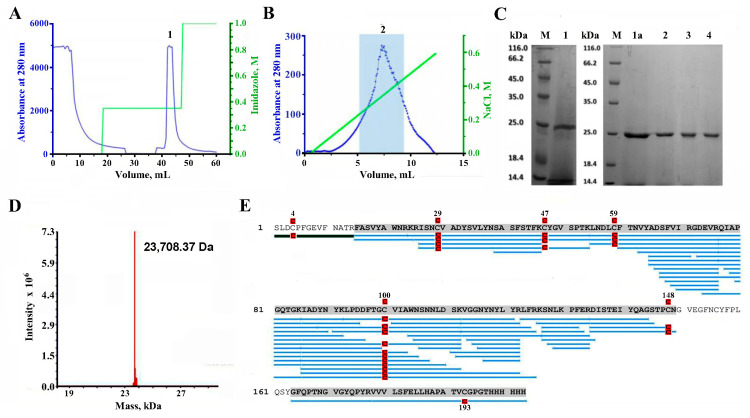
Purification of the recombinant RBD of the Spike-protein of SARS-CoV-2 virus from LEXSY culture medium after 72 h cultivation. (**A**) Affinity metal-chelate chromatography of the produced RBD on Ni-INDIGO agarose resin in stepped imidazole gradient (colored green). Eluted peak 1 corresponds to the target product. (**B**) A cation exchange chromatography elution profile of the collected and dialyzed peak 1 in linear NaCl gradient (colored in green). Fractions from central part of peak 2 (shaded blue) were pooled in a volume of 5–10 mL for subsequent desalting on disposable columns. (**C**) SDS-PAGE of RDB at different stages of purification–the peak 1 after metal-chelate chromatography before (track 1) and after dialysis in 20 mM MES (track 1a), pooled fraction 2 after a cation exchange chromatography before (track 2), and after desalting (tracks 3 and 4). Tracks M–standards with indication of molecular masses. (**D**) ESI MS data for the final RBD sample, indicating its molecular mass. (**E**) Primary structure confirmation for purified RBD by MS analysis of its tryptic digest, visualizing the coverage of the protein sequence. MS2 fragmentation spectral matches covered 86.6% of the protein sequence (shaded in gray). Regions in the protein sequence that are covered by supporting peptides are displayed in blue. Cys-residue modification (carbamidomethylation) is shown by red icon. Peptide containing the putative N-glycosylation site (green line) in MS/MS spectra was detected manually.

**Figure 4 molecules-29-00943-f004:**
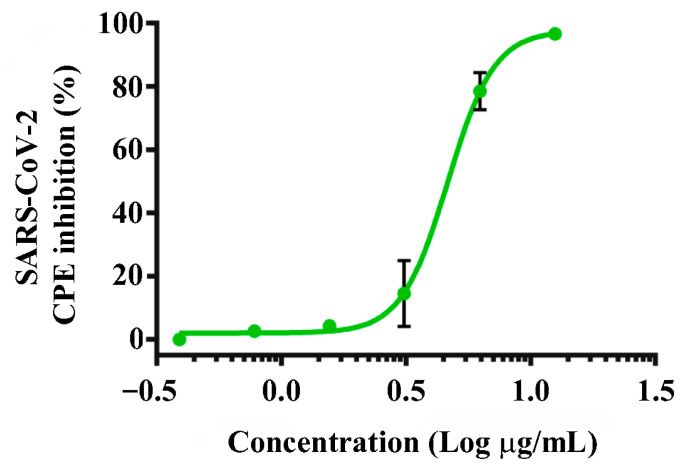
Ability of LEXSY-expressed recombinant RBD to inhibit entry of SARS-CoV-2 virus. Dose-dependent inhibition by RBD protein of a cytopathic effect in Vero E6 cells induced by the SARS-CoV-2 virus. The data are presented as mean inhibition (%) ± s.e.m. (*n* = 3).

**Figure 5 molecules-29-00943-f005:**
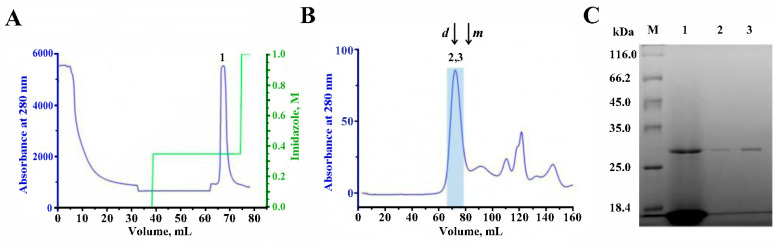
Purification of the recombinant *Lymnaea stagnalis* AChBP from LEXSY culture medium. (**A**) Affinity metal-chelate chromatography of the produced AChBP on Ni^2+^-INDIGO agarose resin in stepped imidazole elution (colored in green). Eluted peak 1 corresponds to the target product. (**B**) Gel-filtration elution profile of the collected peak 1 after metal-chelate chromatography, presumably corresponding to the pentameric form of the *Lymnaea stagnalis* AChBP, on a column with Toyopearl HW-55F resin at a 1 mL/min flow rate. The first peak eluting at 65–75 min (fractions 2 and 3, shaded blue) corresponds to the pentameric form of the target protein with an approximate mass of about 120–130 kDa, according to column calibration with monomeric (m) and dimeric (d) forms of BSA (shown by arrows). (**C**) SDS-PAGE (monomeric form) of the target protein at different stages of purification–the peak 1 after metal-chelate chromatography (track 1), the fractions 2 and 3 after gel-filtration (tracks 2 and 3). Track M–standards with indication of molecular masses.

**Figure 6 molecules-29-00943-f006:**
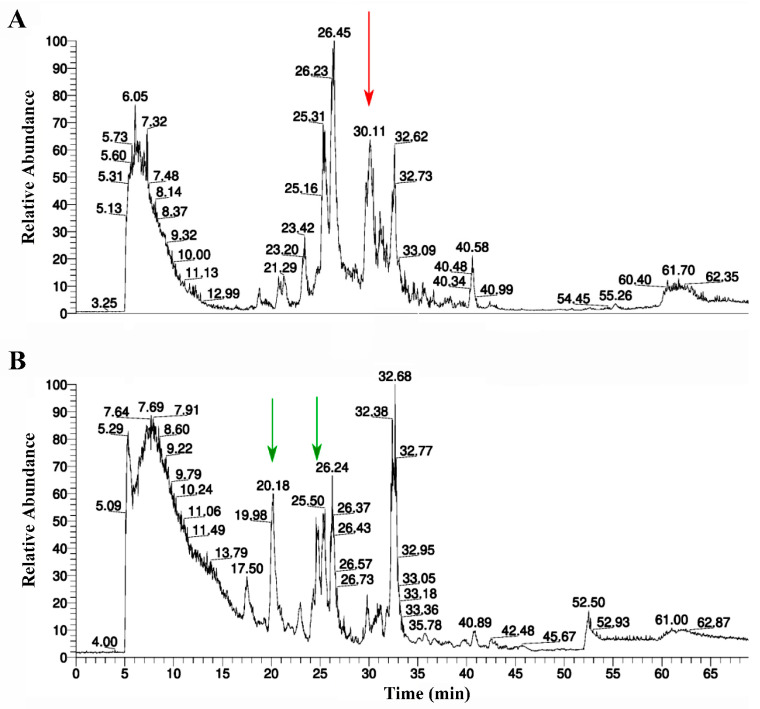
HPLC elution profiles of thermolysin digestion of the recombinant *Lymnaea stagnalis* AChBP purified from LEXSY culture medium. (**A**) Chromatography of the thermolysin-digested protein before the PNGase treatment. Red arrow shows the peak (at 30.1 min), which disappears after sugar cleavage from glycopeptide; (**B**) Chromatography of the thermolysin-digested AChBP after the PNGase treatment. Green arrows show peaks (at 20.2 and 25.5 min) that emerged after the sugar cleavage from glycopeptides.

**Figure 7 molecules-29-00943-f007:**
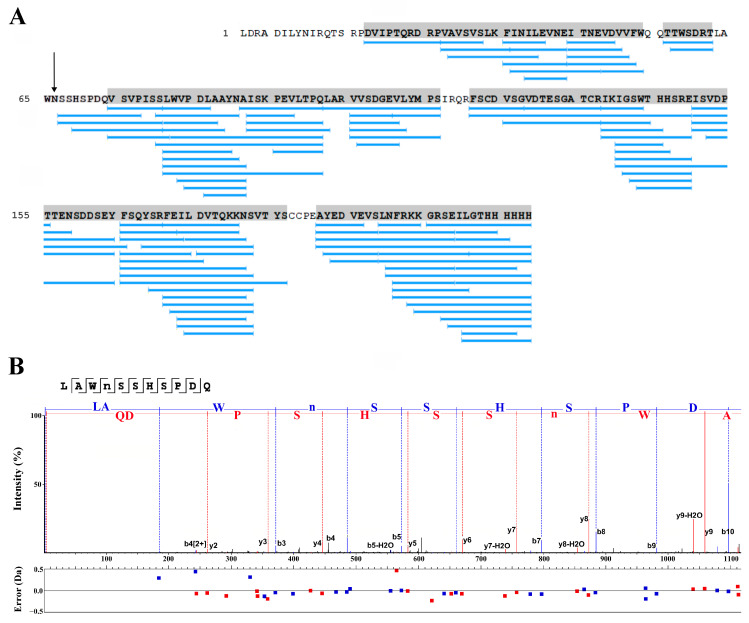
Primary structure confirmation for recombinant *Lymnaea stagnalis* AChBP. (**A**) In the absence of PNGase treatment, about 86.2% of the protein structure was covered in the MS2 fragmentation spectral matches. Blue bars represent spectral matches assigned by two fragment ion series with high confidence. Peptide containing the putative N-glycosylation site was not detected by PEAKS software (arrow). (**B**) MS2 fragmentation assignment of the peptide with putative N-glycosylation site, which had been detected only after the PNGase treatment of the AChBP thermolysin digest. Lower panel demonstrates the absolute error (in Daltons) of the peptide fragment assignment. Blue sequence is reconstructed from b-series ion fragments, red sequence is reconstructed on the basis of y-series ion fragments.

**Figure 8 molecules-29-00943-f008:**
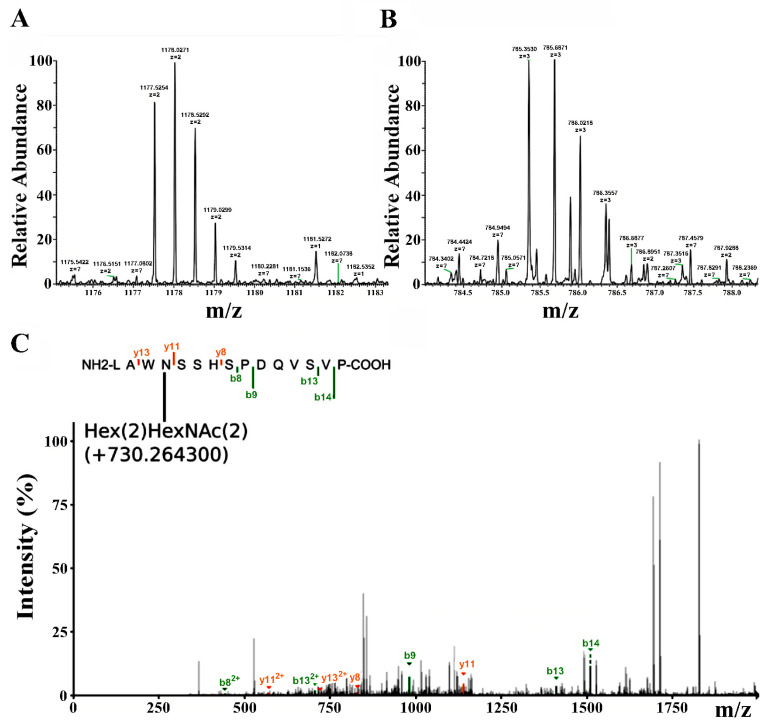
Mass-spectrometry analysis of the peptide fragment modified with glycan. (**A**) Peaks corresponding to isotope distribution consistent with ^63^LAWNSSHSPDQVSVP^77^ peptide modified with glycan bearing double charge. Peaks which charges were not automatically attributed are marked with question marks. (**B**) Peaks corresponding to isotope distribution consistent with ^63^LAWNSSHSPDQVSVP^77^ peptide modified with glycan bearing triple charge. (**C**) MS2 fragmentation assignment of the peptide with N-glycosylation site-bearing core glycan consisting of two hexose and two mannose residues. Several fragment ions were matched to the MS2 spectrum, confirming the peptide structure. The difference in mass was 730.3 Da (shown in parenthesis), corresponding to the glycan. Green denotes b-series fragmentation and red color denotes y-series fragmentation.

**Figure 9 molecules-29-00943-f009:**
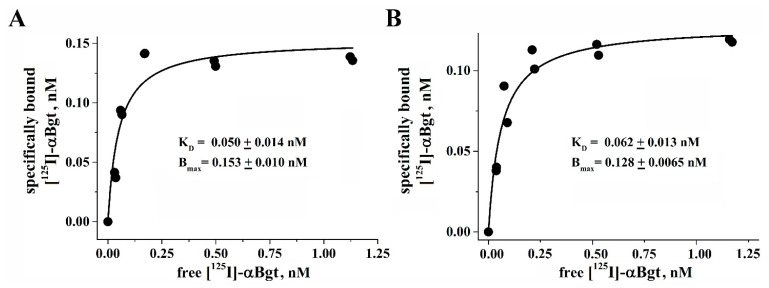
Ligand binding properties of the recombinant LEXSY-produced *Lymnaea stagnalis* AChBP. (**A**) Binding curve of [^125^I]-αBgt to the recombinant *L. stagnalis* AChBP produced in LEXSY. (**B**) Binding curve of [^125^I]-αBgt to the recombinant *L. stagnalis* AChBP produced in the insect cell Sf9.

**Table 1 molecules-29-00943-t001:** Amount and concentration of recombinant RBD at purification stages of isolation.

Stage	Fraction Volume, mL	Concentration (SDS-PAGE Evaluation), μg/mL	Total Protein Amount, μg
crude supernatant	150	ND	ND
affinity chromatography	5	100	500
ion-exchange chromatography	4	85	340
desalting and concentration	1	270 ^1^	270 ^1^

ND—not determined; ^1^—averaged estimate according to the Bradford assay and UV spectrometry.

**Table 2 molecules-29-00943-t002:** Amount and concentration of recombinant AChBP at purification stages of isolation.

Stage	Fraction Volume, mL	Concentration (SDS-PAGE Evaluation), μg/mL	Total Protein Amount, μg
crude supernatant	300	ND	ND
affinity chromatography	5	200	1000
size-exclusion chromatography	10	70	700
concentration	1	400 ^1^	400 ^1^

ND—not determined; ^1^—averaged estimate according to the Bradford assay and UV spectrometry.

## Data Availability

Data are contained within the article and Appendix A.

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
