# Peer review of "Efficient Expression in Leishmania tarentolae (LEXSY) of the Receptor-Binding Domain of the SARS-CoV-2 S-Protein and the Acetylcholine-Binding Protein from Lymnaea stagnalis"

_molecules, 2024, doi:10.3390/molecules29050943_

Round 1
Reviewer 1 Report
Comments and Suggestions for Authors
This manuscript described the functional expression of two types of proteins, the receptor binding domain (RBD) of the spike protein of SARS-CoV-2 virus and homopentameric acetylcholine-binding protein (AChBP) from Lymnaea stagnalis, using a novel heterologous protein expression system based on Leishmania tarentolae, which offers an alternative to yeast/insect/mammalian cells. This system has several advantages and this kind of study will be of interest to some researchers struggling with expression of complex proteins. The manuscript is well written and the most of the data is convincing. However, in recent years the advantages and disadvantages of the Leishmania system have become clear, I feel that it is not much novel to express these two proteins with this system. The RBD of SARS-CoV-2 virus has only the same anti-viral activity as the RBD expressed in E. coli., and has only similar expression efficiency as the RBD expressed in animal cells, despite its relatively small size. As for AChBP from Lymnaea stagnalis, it probably cannot be expressed in the E. coli system, but there does not seem to be much advantage of expressing in the Leishmania system compared to the insect cell system. Authors should not only make these recombinant proteins and evaluate them, but also use these proteins to challenge something novel.
Specific comments
1- In this kind of research, authors should make a purification table. How much is starting culture volume? How much yield was at each purification stage?
2- In figure 5B, authors should show the evidence that the peak is pentamer. At least, authors should add BSA data.
Reviewer 2 Report
Comments and Suggestions for Authors
This paper is surprisingly well-written. I almost love this manuscript. Materials and methods are vigorously written so that everyone can follow them. The analysis was well done with several aspects. This manuscript can be a textbook of LEXSY. I don't have any further comments on this manuscript, but proofreading the manuscript in English can improve it.
Comments on the Quality of English LanguageEnglish proofreading can improve the manuscript.
Reviewer 3 Report
Comments and Suggestions for Authors
The manuscript "Efficient expression in Leishmania tarentolae (LEXSY) of the receptor-binding domain of the SARS-CoV-2 S-protein and the acetylcholine-binding protein from Lymnaea stagnalis" by Son et al provides experimental conditions for the production of glcosylated protein for two target systems: AChBp and the Sars-Cov_2 RBD. The work follows manufacturer's protocol to produce samples that are already reported elsewhere in the literature. The novelty is that LEXSY provides a cheaper mechanism to access glycosylated protein than other systems, but a weakness is that the impact or natural of the glycoslation pattern has not been compared to that expected in the in vivo scenario and further comparisons and experiments are needed prior to publication. There's no clear link between AChBp and RBD, other than that both are produced in the LEXSY system and no data for the need for glycosylation in the samples or the presented analysis is shown.
1. RDB: Figure 4: How does the LEXSY generated material compare to that from RBD generated in mammalian systems. Is there any significant impact on RBD behaviour compared with that from literature or from LEXSY samples after removal of the glycans? (PNGase?). Or from RBP generated by other expression systems.
2. AChBp: LEXSY produced sample is compared to that from another recombinant system (Sf9). Again the impact of glycosylation needs to be underlined by comparison with samples with glycans removed and a comparison of the LEXSY generated glycosylation pattern with that expect in vivo.
Comments on the Quality of English Languagedetailed sample preparation, but no novel experiments or insight on impact of glycosylation on sample behaviour
Reviewer 4 Report
Comments and Suggestions for Authors
In the article ‘Efficient expression in Leishmania tarentolae (LEXSY) of the receptor-binding domain of the SARS-CoV-2 S-protein and the acetylcholine-binding protein from Lymnaea stagnalis’, the authors explore the use of the Leishmania tarentolae (LEXSY) system for expressing the receptor-binding domain of the SARS-CoV-2 Spike protein and the acetylcholine-binding protein from Lymnaea stagnalis. The focus is on evaluating the methodology, results, and broader implications of this approach within the field of protein expression and its potential impact on biomedical research. The article is well written and is suitable for publication after addressing the following concerns.
1. Discussion of LEXSY Expression System Limitations: The article would benefit from a detailed discussion on the limitations or drawbacks of the LEXSY expression system. Including this information would provide a more comprehensive understanding for the readers.
2. Comparative Analysis with Other Expression Methods: It would be valuable to include a comparative analysis of the LEXSY system with other protein expression methods. Emphasizing parameters such as total expression time and yield would offer readers a clearer perspective on the system's relative efficiency.
3. Spin Labeling for NMR Studies: The manuscript should address the challenges or ease associated with spin labeling for NMR studies, like 15N labeling. Insights into the practical aspects of using these proteins in advanced research techniques would be beneficial.
4. Uniformity in Figure Presentation: The use of fonts and font sizes for figure axis labels appears to be inconsistent. Standardizing these elements across all figures would enhance the manuscript's readability.

Comments on the Quality of English LanguageIn some parts of the manuscript, there are minor inconsistencies in English usage. It might be beneficial to have these sections reviewed by a native English speaker to enhance the overall clarity and coherence of the text.
Round 2
Reviewer 1 Report
Comments and Suggestions for Authors
In this revised manuscript, the authors have made changes according to my original review comments. I understand the situation regarding the glycosylation of their target proteins. I am satisfied with their responses and in favor of acceptation of this manuscript at current form for publication in Molecules.